# An Observational Study on the Prediction of Range of Motion in Soldiers Diagnosed with Patellar Tendinopathy Using Ultrasound Shear Wave Elastography

**DOI:** 10.3390/bioengineering11121263

**Published:** 2024-12-13

**Authors:** Min-Woo Kim, Dong-Ha Lee, Young-Chae Seo

**Affiliations:** 1Department of Orthopedic Surgery, Busan Medical Center, Busan 47527, Republic of Korea; drkimminwoo@naver.com; 2Department of Orthopaedic Surgery, 5th Air Mobility Wing, Republic of Korea Air Force Military Orthopaedic Surgeon, Busan 46718, Republic of Korea; osint4@naver.com

**Keywords:** patellar tendon injuries, ultrasonography, elasticity imaging techniques, range of motion, articular, rehabilitation, logistic models

## Abstract

Introduction: This study hypothesized that changes in the elasticity of the quadriceps and patellar tendons before and after the diagnosis of patellar tendinopathy would correlate with the range of motion (ROM) following conservative treatment. We aimed to prospectively assess post-treatment ROM using multinomial logistic regression, incorporating elasticity measurements obtained via shear wave elastography (SWE). Materials and Methods: From March 2023 to April 2024, 95 patients (86 men; aged 20–45 years, mean 25.62 ± 5.49 years) underwent SWE preoperatively and two days post-diagnosis of patellar tendinopathy. Elasticity measurements of the rectus femoris, vastus medialis, vastus lateralis, patellar tendon, and biceps tendon were obtained during full flexion and extension. Based on ROM 56 days post-treatment, patients were categorized into two groups: Group A (ROM > 120 degrees) and Group B (ROM < 120 degrees). A multinomial logistic regression algorithm was employed to classify the groups using patient information and tendon elasticity measurements both at diagnosis and 1-week post-diagnosis. Results: The predictive accuracy using only patient information was 62%, while using only elasticity measurements yielded 68% accuracy. When combining patient information with elasticity measurements taken at diagnosis and two days post-diagnosis, the algorithm achieved an accuracy of 79%, sensitivity of 92%, and specificity of 56%. Conclusions: The combination of patient information and tendon elasticity measurements obtained via SWE at pre-conservative treatment and early post-conservative treatment periods effectively predicts post-treatment ROM. This algorithm can guide rehabilitation strategies for soldiers with patellar tendinopathy.

## 1. Introduction

Shear Wave Elastography (SWE) is a novel ultrasound imaging technology capable of measuring tissue elasticity in the human body [1,2,3]. SWE has been widely adopted and studied in fields such as breast, liver, and thyroid imaging. In breast imaging, for example, SWE has advanced to the point where it can significantly distinguish between benign and malignant masses by analyzing tissue elasticity.

Building on this technology, various ultrasound examinations are now being conducted in the musculoskeletal system. Previous studies have analyzed tissue elasticity, such as tendons, using strain elastography, and a precursor to SWE [1,4,5]. However, strain elastography relies on the force applied by the operator with the probe, leading to variability in results and challenges in securing reliable measurements. SWE addresses these limitations by reducing operator-dependent variability. Despite its potential, few studies have utilized SWE to analyze musculoskeletal tissue elasticity.

One of the remarkable advancements SWE has brought to musculoskeletal imaging is its ability to quantitatively assess tissue stiffness, a parameter directly related to tissue pathology [6,7]. In the context of patellar tendinopathy, SWE provides a unique opportunity to measure and monitor changes in the elasticity of the patellar tendon over time [8]. Unlike conventional ultrasound imaging, which primarily captures structural details, SWE adds a functional dimension, allowing clinicians to gauge subtle differences in tendon stiffness that might signal early degeneration or response to treatment [4]. By characterizing the mechanical properties of tendons and other soft tissues, SWE helps address an unmet need in diagnosing and monitoring musculoskeletal disorders where conventional imaging has limitations [9]. Thus, the clinical value of SWE in patellar tendinopathy may lie in its potential as a non-invasive biomarker of tendon health and function, aiding both in initial diagnosis and in tracking recovery progress during rehabilitation [10].

Patellar tendinopathy [11], an overuse injury primarily affecting athletes, has been a research focus for nearly five decades, reflecting the ongoing quest for deeper pathological understanding and improved clinical interventions [12,13]. Described initially by Dr. Martin Blazina in 1973 as “jumper’s knee”, this condition involves pain at the quadriceps insertion (25% of cases), the distal pole of the patella (65%), and the tibial tubercle (10%) [14]. Specifically, patellar tendinopathy refers to enthesiopathy at the distal pole of the patella [15]. Despite being termed tendinitis initially, histological analysis reveals a lack of inflammation.

Proximal patellar tendinopathy is influenced by biomechanical factors involving the knee extensor mechanism, which includes the quadriceps muscle and tendon, the patella, and the patellar tendon [16]. These structures play a pivotal role in dynamically stabilizing the knee during valgus and rotatory stresses. Activities such as jumping and landing impose significant eccentric loads on the patellar tendon as the knee controls the braking of the lower limb on the ground. Repeated exposure to these stresses can contribute to tendon overload and subsequent injury, underscoring the importance of addressing these factors in rehabilitation strategies.

The prognosis for patellar tendinopathy is often poor. Kettunen’s 15-year study of 36 athletes found that 53% of those with patellar tendinopathy had to retire from sports, compared to 7% of control athletes [17,18]. Non-operative treatments, including physiotherapy, load management, and injections, are primary, though roughly 10% of patients require surgery [19].

Given these challenges, it is crucial to develop tools for predicting and managing range of motion (ROM) limitations in patellar tendinopathy. Changes in tendon elasticity, measurable by SWE, can provide insights into post-treatment ROM outcomes. Combining patient data with SWE-derived elasticity metrics at diagnosis and shortly after can enhance prediction accuracy, offering a valuable tool for clinicians. This approach aims to improve rehabilitation strategies, ultimately enhancing patient outcomes and guiding future research in this field. B-mode ultrasound typically reveals hypoechoic thickening of the proximal and deep fibers of the patellar tendon, along with cortical pitting at the inferior pole of the patella [20]. Color and power Doppler imaging often demonstrate vascular invasion of the tendon tissue, originating from vessels within Hoffa’s fat pad. These findings emphasize the structural changes in the tendon that may contribute to ROM limitations and highlight the potential of SWE to quantify these changes and provide objective data for clinical decision-making.

Despite its advantages, SWE application in musculoskeletal disorders, particularly in dynamic and load-bearing tendons such as the patellar tendon, presents some challenges. Factors such as patient positioning, the angle of the probe, and the specific region of the tendon being measured can influence SWE results, necessitating standardized protocols for reliable data acquisition. Moreover, differences in SWE technology between ultrasound manufacturers and devices may impact elasticity values, underscoring the need for consistency and calibration in clinical settings. To fully integrate SWE into routine musculoskeletal practice, further studies are essential to establish reference values for tendon elasticity, particularly in active populations at risk for tendinopathy. Establishing normative data and refining technique protocols will be critical for leveraging SWE as a predictive tool for a range of motion limitations and treatment outcomes in patellar tendinopathy.

Factors affecting ROM include the quadriceps muscles, quadriceps tendon, and patellar tendon [21]. These structures are also influenced by surgical interventions. Several studies using SWE are being conducted to objectively assess these effects. However, research linking changes in tissue elasticity to ROM after the diagnosis of patellar tendinopathy remains limited. This study aimed to determine if post-treatment ROM could be predicted based on patient history (age, sex, HTN, DM), ROM at diagnosis, and the elasticity of the quadriceps muscle/tendon and patellar tendon before and one week after treatment. Specifically, we measured the elasticity of the quadriceps muscle, quadriceps tendon, and patellar tendons using SWE. Based on these measurements, we predicted post-treatment ROM using a multinomial logistic regression method.

## 2. Material and Methods

### 2.1. Patients

This study was approved by the Institutional Review Board (Approval #P01-202304-11-002) of the Public Institutional Bioethics Committee designated by the Ministry of Health and Wellness and written informed consent was obtained from all the patients. The patients provided their consent for publication of the data concerning the case. All diagnoses were performed by two experienced orthopedic surgeons. A total of 95 patients (86 [90.5%] males and 9 [9.5%] females) with state after, were enrolled in this clinical study. This resulted in 475 post-treatment tendons (95 patellar and 76 quadriceps) and 475 pre-treatment tendons (50 patellar and 50 quadriceps) for analysis. The diagnosis of proximal patellar tendinopathy was established based on the presence of tenderness upon palpation of the proximal patellar tendon at the inferior pole of the patella, limitation of knee flexion range of motion to 120 degrees or less, and pain elicited during a single-leg squat, which places a significant load on the patellar tendon. These clinical parameters ensured a consistent and standardized approach for identifying patients with this condition.

### 2.2. Exclusion Criteria

Patients with open wounds, acute infections, fractures around the knee, or any comorbidities that prevented a clinical and sonographic evaluation of the knee joint were excluded from this study.

### 2.3. Conservative Treatment Protocol

The knee was kept in full extension with a functional brace for three weeks following conservative treatment. During this period, patients were allowed to perform passive range of motion (ROM) exercises under supervision to maintain joint mobility without placing undue stress on the tendon. Gradual and partial motion restoration began in the fourth week after treatment, with full weight bearing permitted starting after the third week. This protocol was designed to balance the protection of the healing tendon with the prevention of joint stiffness and muscle atrophy [11,12,22]. Patients were given 200 mg of Celebrex (celecoxib; Pfizer, New York, NY, USA) as premedication, taken once daily (qd). Additionally, patients received Esogen Tablet 20 mg (Esomeprazole Magnesium, ALVOGEN, Gwangju, Republic of Korea) once daily (qd).

### 2.4. Clinical Assessment

Every participant underwent a standardized clinical examination by two orthopedic specialists. During the clinical visit, all patients completed a standardized questionnaire concerning their general medical history, current pain medication, number of knee surgeries, and history of any knee infection. The range of motion was measured at pre-treatment and at 1, 4, and 8 weeks after treatment.

### 2.5. Elasticity Measurement by Shear Wave Elastography (SWE)

Ultrasound examinations were performed on patients diagnosed with patella tendinopathy at the 5th Air Force Mobility Wing Medical Battalion. Tissue elasticity was measured using SWE both pre-treatment and one week post-treatment. The elasticity of the rectus femoris, vastus medialis, vastus lateralis, patella tendon, and lateral biceps tendon were assessed during maximum flexion and extension. A 5–12 MHz linear probe (LOGIQ e, GE, Boston, MA, USA) was utilized. Measurements were taken where the Reliability Measurement Index (RMI) was above 0.7, ensuring high reliability and accuracy according to the manufacturer’s guidelines and the existing literature [23].

### 2.6. ROM Prediction

From the medical records of the patients, HTN, DM, preoperative ROM, age, and gender were recorded. The elasticity of a total of 20 tissues was measured. The elasticity of rectus femoris, vastus medialis, vastus lateralis, patella tendon, and lateral biceps tendon at maximum flexion and extension was measured for the pre-treatment and post-treatment period (1 week after diagnosis of patella tendinopathy). To predict the final post-treatment ROM, multinomial logistic regression was performed using a total of 25 parameters as inputs (Figure 1).

In this paper, all patients were classified as Group A (ROM > 120 degrees) and Group B (ROM < 120 degrees). Accuracy, sensitivity, and specificity were calculated to measure the performance of the predictive model. In addition, to understand the importance of SWE in this prediction model, multinomial logistic regression was performed to predict postoperative ROM only with patient information. As before, accuracy, sensitivity, and specificity were measured to see how much difference was made.

Logistic regression (LR) predicts the probability that each patient belongs to either group A or B and classifies the patient through the probability value and threshold [24]. Specifically, LR computes the log-odds using a linear combination as $z_{j}=w_{0}+\sum_{i=1}^{25}w_{i}x_{ij}$ where $z_{j}$ is the log-odds of the jth patient, $w_{0}$ is the intercept, $w_{i}$ is the ith coefficient, and $x_{ij}$ is the ith feature of the jth patient. Then, LR uses the sigmoid function $f_{sig}(\cdot )$ to provide the probability value ranging from 0 to 1 from the log odd. Through the cross-entropy loss function, LR finds optimal coefficients as $\left\{{\hat{w}}_{0},\ldots ,\;{\hat{w}}_{28}\right\}=\underset{\left\{w_{0},\ldots ,\;w_{28}\right\}}{\mathrm{argmin}}-\frac{1}{N}\sum_{j=1}^{N}\left[y_{j}log(f_{sig}\left(z_{j}\right))+(1-y_{j})log(1-f_{sig}\left(z_{j}\right))\right]$ where $y_{j}$ denotes the target group of the jth patient (Group A = 1, Group B = 0), and N is the number of patients. Finally, LR estimates the probability for every patient using the coefficients as $p_{j}={f_{sig}(\hat{w}}_{0}+\sum_{i=1}^{25}{\hat{w}}_{i}x_{ij})$. The final step is classifying the patient using a threshold θ=0.5 so that if $p_{j}>\theta$, the class is Group A, otherwise, the class is Group B. Additionally, by changing threshold, ROC curve and optimal threshold were made. In this study, we employed ‘mnrit’ and ‘mnrval’ functions in MATLAB for the LR processes.

### 2.7. Statistical Methods

Data analysis was performed using IBM SPSS Statistics version 22.0. Continuous variables were expressed as mean ± standard deviation (SD) or median values with interquartile ranges (IQRs) and compared using independent Student’s *t*-tests. Categorical variables were presented as numbers and percentages and analyzed with the Chi-square test or Fisher’s exact test, as appropriate. The sample size was determined using G*Power software (3.1 9.7.) to ensure sufficient statistical power, targeting a medium effect size (Cohen’s d = 0.5).

## 3. Results

### 3.1. Patient Characteristics

The overall demographics are shown in Table 1. A total of 95 patients, with a majority being male and a small proportion female, were included in the final analysis. The mean age of the patients was mid-twenties. There was no statistical significance between Group A and Group B.

### 3.2. Prediction of ROM

To predict the range of motion (ROM) at 4 weeks and 8 weeks after the diagnosis of patella tendinopathy, three models were used:A model using only the patient’s basic information and preoperative ROM.A model using only the SWE values pre-treatment and 1-week post-treatment.A model combining all values from both the patient’s basic information, preoperative ROM, and SWE values.

The accuracy, sensitivity, and specificity of these models are shown in Table 2. The model using only SWE values performed better in terms of accuracy, sensitivity, and specificity compared to the model using only the patient’s basic information and preoperative ROM. Additionally, when evaluating other performance metrics, such as the F1-score and Matthews Correlation Coefficient (MCC), the combined model (using both SWE values and patient data) outperformed the other models, further emphasizing the advantage of integrating multiple data sources. These findings highlight the importance of considering various evaluation metrics beyond accuracy alone to better capture the model’s predictive capabilities and overall performance.

To objectively measure the performance of these models, receiver operating characteristic (ROC) curves were calculated The ROC curve was created by calculating sensitivity and specificity while gradually changing the threshold for each model. Based on the ROC curve, the area under the ROC curve (AUC) was calculated [25,26]. Consistent with previous results, the AUC value was higher for the model using only SWE values than for the model using only patient information at both 28- and 56-days post-diagnosis of patella tendinopathy (Figure 2 and Figure 3). Additionally, the model combining patient information with SWE values had the highest AUC value, indicating the most accurate prediction.

## 4. Discussion

In this study, utilizing SWE on quadriceps and patellar tendons following conservative treatment for patella tendinopathy, we demonstrated the significance of SWE in predicting the ROM after treatment. Recent research has shown that SWE reflects changes during tendon healing and reliably measures and displays these changes, aligning with previous studies.

In recent years, SWE has been increasingly employed to better understand the mechanical properties of tendons and to monitor the healing process in tendinopathies. Excellent interobserver, intraobserver, and retest reliability for the quadriceps and patellar tendons have been confirmed. Dirrichs et al. demonstrated in a double-blinded study that this method is superior to B-mode ultrasound (B-US) and power Doppler ultrasound (PD-US) for evaluating tendinopathies [27].

Patellar tendinopathy is characterized by microtears in the tendon, local mucoid degeneration, and fibrinoid necrosis, disrupting the transition from fibrocartilage to mineralized fibrocartilage [12,13,28]. This results in the separation of parallel collagen bundles by mucoid ground substance, leading to a loss of alignment. Increased fibroblast proliferation and neovascularization are observed but without inflammatory cells or elevated prostaglandin levels.

Clinically, patellar tendinopathy presents as anterior knee pain and tenderness localized at the distal pole of the patella. Diagnosis is primarily clinical but supported by imaging findings: ultrasound reveals tendon thickening and heterogeneous echogenicity, while MRI shows blurry ligamentous margins and increased signal intensity within the patellar tendon. X-rays, while useful for excluding other conditions like Osgood-Schlatter disease and Sinding-Larsen-Johansson disease, do not provide specific signs of patellar tendinopathy [29,30].

In this study, the elasticity of the quadriceps and patellar tendons was measured at the time of patella tendinopathy diagnosis and one week later, using SWE. Combining these measurements with the patient’s medical history and preoperative ROM, we predicted ROM at 4 weeks and 8 weeks using multinomial logistic regression. It was shown that SWE is useful in predicting postoperative ROM in patients.

Unlike typical studies that analyze how each factor individually affects ROM, this study assessed whether each factor has a complex effect. It was found that the overall quadriceps tendon values, where no single variable was crucial, impacted ROM. Furthermore, the sensitivity, specificity, and accuracy results from predicting postoperative ROM using (1) commonly known factors and (2) quadriceps elasticity measured by SWE indicated that the latter provided better results. This confirms that SWE measurements can help predict ROM after treatment for patellar tendinopathy. More accurate predictions can be achieved by combining the models presented in this study with existing factors.

The results of this study can aid in predicting ROM two months after conservative treatment. This can facilitate early consideration of active intervention in postoperative care and rehabilitation planning. By combining patient information with elasticity measured by SWE diagnosis and early post-conservative treatment, we can effectively predict postoperative ROM performance. This algorithm can provide valuable direction for rehabilitation.

However, this study has several limitations that should be acknowledged. First, the cohort predominantly consisted of young male soldiers, with only nine female participants (9.5%). This limits the ability to explore potential sex-based differences in tendon elasticity and ROM outcomes, reducing the generalizability of the findings to broader populations. Second, differences in SWE technology and calibration across ultrasound devices may influence elasticity values, raising concerns about reproducibility and consistency in clinical settings. Third, the follow-up duration of 56 days may not fully capture long-term changes in ROM or the sustained effects of tendon elasticity changes post-treatment. Fourth, factors such as patient positioning, probe angle, and measurement site could influence SWE results, introducing variability. The absence of established reference ranges for tendon elasticity in similar populations further restricts the clinical interpretation and applicability of SWE measurements. Fifth, the combined model demonstrated high sensitivity but relatively low specificity, indicating a potential for false-positive predictions. Lastly, potential confounding factors, such as the severity of tendinopathy, variations in rehabilitation protocols, and adherence to prescribed treatments, were not explicitly analyzed.

Future research should focus on addressing these limitations to improve the robustness and clinical relevance of the findings. Specifically, efforts should be made to include more diverse cohorts, ensuring adequate representation of female participants and older populations. Standardized calibration protocols across SWE devices should be developed to minimize variability and improve reproducibility. Longer follow-up periods are needed to evaluate the sustained effects of tendon elasticity changes in ROM outcomes. Moreover, future studies should aim to develop and validate population-specific reference ranges for tendon elasticity to enhance the clinical applicability of SWE measurements. Investigations into confounding factors, such as rehabilitation adherence and treatment protocol variations, should also be incorporated into study designs. Lastly, improving the specificity of the combined model through algorithm refinement and exploring additional metrics, such as MCC and F1-score, will further enhance the predictive accuracy of SWE-based methods.

This study’s findings underscore the potential of SWE as a prognostic tool in managing patellar tendinopathy, particularly by enabling early prediction of post-treatment functional outcomes. By quantifying the elasticity of the quadriceps and patellar tendons, clinicians can identify patients at risk for limited ROM recovery, allowing for the timely modification of treatment plans. The ability to anticipate ROM limitations through non-invasive SWE measurements aligns with the push toward precision medicine, where interventions are increasingly tailored to individual patient profiles. Additionally, the combined use of SWE and conventional clinical factors enhances the robustness of outcome predictions, bridging the gap between diagnosis and rehabilitation planning. These results suggest that incorporating SWE measurements into routine practice could lead to more personalized, targeted approaches in conservative treatment and potentially reduce the likelihood of chronic impairments. Further research and clinical trials could expand these findings, potentially positioning SWE as a standard assessment tool for a broader range of tendinopathies.

## 5. Conclusions

By integrating patient information with elasticity measured by SWE at pre-conservative treatment and early post-conservative treatment periods, we can effectively predict the performance of post-treatment ROM. This algorithm has the potential to guide rehabilitation planning and optimize patient outcomes.

## Figures and Tables

**Figure 1 bioengineering-11-01263-f001:**
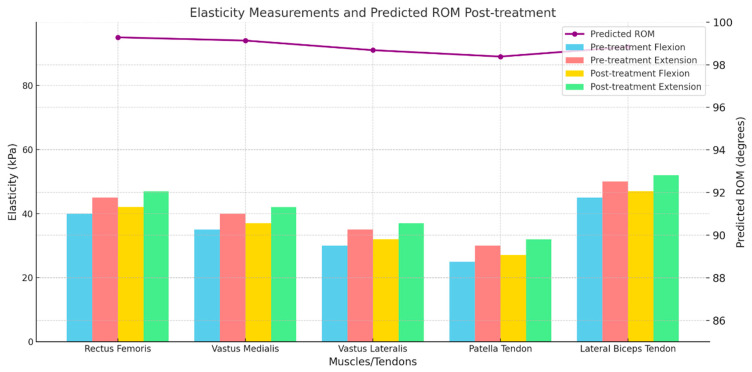
Elasticity Measurements and Predicted ROM of Post-treatment of Patella Tendinopathy.

**Figure 2 bioengineering-11-01263-f002:**
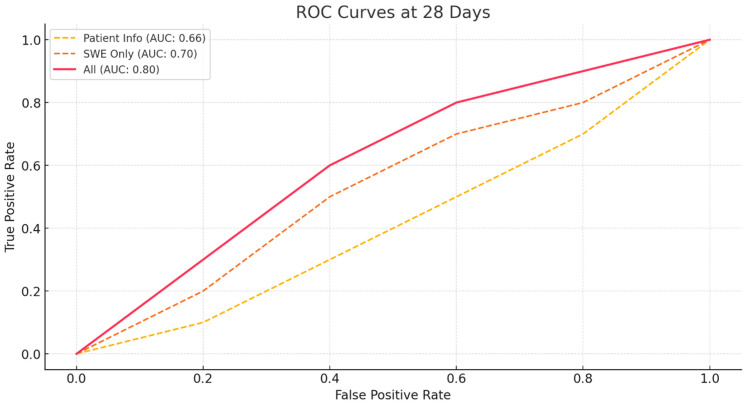
ROC Curves for ROM Prediction at 28 Days Post-Diagnosis Using Different Models.

**Figure 3 bioengineering-11-01263-f003:**
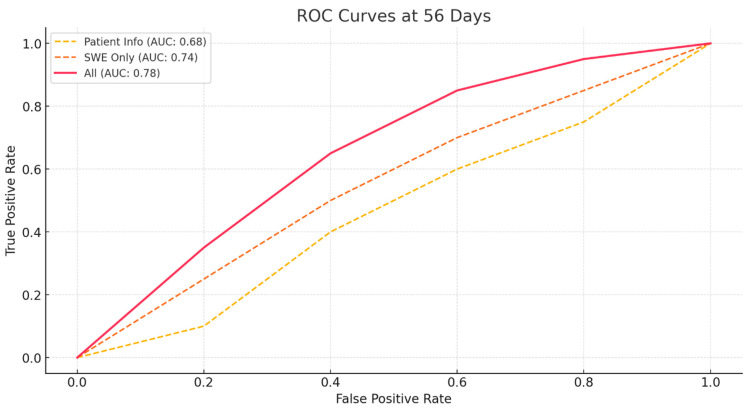
ROC Curves for ROM Prediction at 56 Days Post-Diagnosis Using Different Models.

**Table 1 bioengineering-11-01263-t001:** Demographic data of patients.

	Total	Group A (ROM > 120)	Group B (ROM < 120)
Case (number)	95	61	34
Mean age (years)	24.62 ± 3.49	24.88 ± 3.55	25.14 ± 4.02
Sex (Male/female)	86/9	56/5	30/4
Average ROM	122.79 ± 12.90	123.69 ± 12.11	121.17 ± 13.88

**Table 2 bioengineering-11-01263-t002:** Performance and AUC Values for ROM Prediction at 4- and 8-Weeks Post-Diagnosis of Patellar Tendinopathy Using Models Incorporating Patient Basic Information and SWE Values.

Model	Accuracy (28 Days)	Sensitivity (28 Days)	Specificity (28 Days)	AUC (28 Days)	Accuracy (56 Days)	Sensitivity (56 Days)	Specificity (56 Days)	AUC (56 Days)
Patient Basic Info	0.58	0.42	0.72	0.66	0.62	0.82	0.26	0.68
SWE Only	0.65625	0.56	0.74	0.7	0.68	0.84	0.41	0.74
All	0.76	0.72	0.79	0.8	0.79	0.92	0.56	0.78

## Data Availability

The original contributions presented in this study are included in the article. Further inquiries can be directed to the corresponding author.

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
