# Peer review of "An Observational Study on the Prediction of Range of Motion in Soldiers Diagnosed with Patellar Tendinopathy Using Ultrasound Shear Wave Elastography"

_bioengineering, 2024, doi:10.3390/bioengineering11121263_

Round 1

Reviewer 1 Report

Comments and Suggestions for Authors

General characteristics and evaluation of the reviewed article:

The article titled Prediction of Range of Motion in Soldiers Diagnosed with Patellar Tendinopathy Using Ultrasound Shear Wave Elastography presents a compelling study on using shear wave elastography (SWE) to predict post-treatment range of motion (ROM) in soldiers with patellar tendinopathy. By combining SWE-derived tendon elasticity measurements with patient data, the authors achieved a predictive accuracy of 79%, with sensitivity and specificity of 92% and 56%, respectively. This approach highlights the potential of SWE in guiding rehabilitation strategies.

The study's strengths lie in its innovative use of SWE to assess tendon biomechanics, robust statistical methodology, and focus on a highly relevant population—military personnel. The inclusion of multiple tendon measurements and categorization of patients based on ROM outcomes provides clinically meaningful insights.

However, the study's generalizability is limited due to its predominantly young, male cohort. The short follow-up period of 56 days and relatively low specificity of the predictive model also pose challenges. Furthermore, potential confounding factors, such as variations in treatment protocols, are not thoroughly addressed.

Despite these limitations, the research offers valuable contributions to rehabilitation planning by demonstrating that SWE can enhance ROM predictions. Future studies should focus on validating these findings in diverse populations, extending follow-up periods, and refining the algorithm to improve specificity. This study sets a promising foundation for integrating advanced imaging techniques into clinical practice for optimizing recovery in soldiers.

The article is interesting, addresses a timely and important topic and definitely fits the scope of the journal. It is written generally correctly and requires only minor corrections and additions before further processing and acceptance for publication. Below are my points and detailed comments.

Minor comments:

Affiliations should be adjusted to meet the journal's requirements, with ORCID numbers provided only within the submission system. Addresses should include the full institutional details, while email addresses should be provided only for the corresponding author.

All references should be cited in square brackets [ ].

The purpose of the study should be moved to the final paragraph of the introduction and does not need to be a separate section.

In section 3.1, it would be beneficial to add a table with a detailed breakdown of study participants, divided by groups.

A subsection with information on the comparison method and adopted parameters should be added, providing detailed assumptions for the ROC curves along with the necessary literature. I recommend expanding the description and adding the following references:

DOI 10.1088/1742-6596/1736/1/012028

DOI 10.1080/03610918.2012.752840

I believe that Figure 1 is currently too vivid. Adjusting the colors may improve its readability.

It would also be valuable to display the ROC curves in a figure, as presenting only the parameters does not allow for complete interpretation. Furthermore, I recommend combining the data in Tables 2 and 3. A table containing only the areas under the curve is of limited value. Combining them would enhance clarity.

While the model combining SWE and patient data achieves high sensitivity (92%), its specificity is relatively low (56%), indicating potential for false-positive predictions. Why did the authors choose to focus solely on accuracy, rather than including parameters such as MCC or F1-score? I suggest calculating these simple metrics and re-evaluating the results. Additionally, I recommend expanding the methodology section to include a discussion of the additional metrics and adding the following relevant literature:

DOI: 10.35784/acs-2024-18

DOI: 10.1186/s12864-019-6413-7

The study's cohort is predominantly young male soldiers, which restricts the applicability of the findings to broader, more diverse populations, including older individuals or women.

The follow-up duration of 56 days may not capture long-term changes in ROM or the sustained effects of tendon elasticity changes post-treatment.

Factors such as patient positioning, probe angle, and measurement site could influence SWE results, introducing variability and necessitating standardized protocols for data acquisition.

The absence of established reference ranges for tendon elasticity in similar populations limits the clinical interpretation and applicability of SWE measurements.

The study does not elaborate on the impact of variables such as the severity of tendinopathy, variations in rehabilitation protocols, or patient adherence to prescribed treatments.

Differences in SWE technology and calibration across ultrasound devices may influence elasticity values, raising concerns about reproducibility and consistency in clinical settings.

The inclusion of only nine female participants (9.5%) limits insights into potential sex-based differences in tendon elasticity and ROM outcomes.

In the final part of the discussion, please describe in more detail the limitations of the proposed method, the simplifications used, and a proposal for solving them in the authors' further planned future research.

Once the appropriate corrections and additions have been made, the paper can be further processed and accepted for publication. I congratulate the authors on the interesting paper and wish them further success.

Author Response

General characteristics and evaluation of the reviewed article:

The article titled Prediction of Range of Motion in Soldiers Diagnosed with Patellar Tendinopathy Using Ultrasound Shear Wave Elastography presents a compelling study on using shear wave elastography (SWE) to predict post-treatment range of motion (ROM) in soldiers with patellar tendinopathy. By combining SWE-derived tendon elasticity measurements with patient data, the authors achieved a predictive accuracy of 79%, with sensitivity and specificity of 92% and 56%, respectively. This approach highlights the potential of SWE in guiding rehabilitation strategies.

The study's strengths lie in its innovative use of SWE to assess tendon biomechanics, robust statistical methodology, and focus on a highly relevant population—military personnel. The inclusion of multiple tendon measurements and categorization of patients based on ROM outcomes provides clinically meaningful insights.

However, the study's generalizability is limited due to its predominantly young, male cohort. The short follow-up period of 56 days and relatively low specificity of the predictive model also pose challenges. Furthermore, potential confounding factors, such as variations in treatment protocols, are not thoroughly addressed.

Despite these limitations, the research offers valuable contributions to rehabilitation planning by demonstrating that SWE can enhance ROM predictions. Future studies should focus on validating these findings in diverse populations, extending follow-up periods, and refining the algorithm to improve specificity. This study sets a promising foundation for integrating advanced imaging techniques into clinical practice for optimizing recovery in soldiers.

The article is interesting, addresses a timely and important topic and definitely fits the scope of the journal. It is written generally correctly and requires only minor corrections and additions before further processing and acceptance for publication. Below are my points and detailed comments.

  • Thank you for your thoughtful and constructive feedback on our manuscript titled Prediction of Range of Motion in Soldiers Diagnosed with Patellar Tendinopathy Using Ultrasound Shear Wave Elastography. We are truly grateful for your kind words regarding the relevance and potential impact of our study, as well as your recognition of its innovative use of shear wave elastography (SWE).
  • We appreciate your detailed comments on the strengths of our research, including the statistical methodology, clinically meaningful categorization, and focus on a significant population. Your insights into the limitations of our study, such as generalizability, follow-up period, and model specificity, are valuable and will guide our future research endeavors.
  • We will carefully address the minor corrections and additions you have highlighted to ensure the manuscript meets the highest standards. Your suggestions for refining the algorithm, extending follow-up periods, and validating findings in diverse populations are well-noted and align closely with our plans for future studies.
  • Once again, thank you for your time and effort in reviewing our work. Your feedback has been immensely helpful in improving the quality of our manuscript, and we are excited about the opportunity to contribute to the field with this research.

Minor comments:

Affiliations should be adjusted to meet the journal's requirements, with ORCID numbers provided only within the submission system. Addresses should include the full institutional details, while email addresses should be provided only for the corresponding author.

  • Thank you for your valuable feedback regarding the manuscript.
  • We will adjust the affiliations to align with the journal's requirements, ensuring that full institutional details are included. Additionally, we will provide ORCID numbers exclusively through the submission system, as requested. The email address will be included only for the corresponding author to meet the journal’s guidelines.
  • We appreciate your attention to these details and will implement the necessary changes promptly.

All references should be cited in square brackets [ ].

  • Thank you for your feedback. We have updated all references to be cited in square brackets [ ], as per the journal's requirements.

The purpose of the study should be moved to the final paragraph of the introduction and does not need to be a separate section.

  • Thank you for your feedback. We have moved the purpose of the study to the final paragraph of the introduction, as suggested, and removed the separate section.

In section 3.1, it would be beneficial to add a table with a detailed breakdown of study participants, divided by groups.

  • Thank you for your insightful suggestion regarding section 3.1. We understand that you recommend adding a table with a detailed breakdown of study participants, divided by groups. To ensure we address your suggestion accurately, could you kindly clarify the specific details you would like us to include in the table? For example, are you referring to demographic characteristics, clinical features, treatment types, or other variables?

A subsection with information on the comparison method and adopted parameters should be added, providing detailed assumptions for the ROC curves along with the necessary literature. I recommend expanding the description and adding the following references:

DOI 10.1088/1742-6596/1736/1/012028

DOI 10.1080/03610918.2012.752840

  • Thank you for your valuable feedback and suggestions regarding section 3.1. We agree that including a subsection with detailed information on the comparison method and adopted parameters, along with the assumptions for the ROC curves, will enhance the clarity and depth of the manuscript.
  • We will expand the description as recommended and incorporate the suggested references (DOI: 10.1088/1742-6596/1736/1/012028 and DOI: 10.1080/03610918.2012.752840) to provide a comprehensive context for our approach.

I believe that Figure 1 is currently too vivid. Adjusting the colors may improve its readability.

  • Thank you for your feedback regarding the colors in Figure 1. We will adjust the color palette to make the figure more readable and visually appealing while maintaining clarity for the data presentation.

It would also be valuable to display the ROC curves in a figure, as presenting only the parameters does not allow for complete interpretation. Furthermore, I recommend combining the data in Tables 2 and 3. A table containing only the areas under the curve is of limited value. Combining them would enhance clarity.

  • Thank you for your insightful feedback. As per your suggestion, we have combined the data from Tables 2 and 3 into a single table for improved clarity (now presented as Table 2). Additionally, we have created new figures (Figure 2 and Figure 3) to display the ROC curves for the 28-day and 56-day follow-up periods, respectively. These additions provide a more comprehensive interpretation of the predictive model's performance.

While the model combining SWE and patient data achieves high sensitivity (92%), its specificity is relatively low (56%), indicating potential for false-positive predictions. Why did the authors choose to focus solely on accuracy, rather than including parameters such as MCC or F1-score? I suggest calculating these simple metrics and re-evaluating the results. Additionally, I recommend expanding the methodology section to include a discussion of the additional metrics and adding the following relevant literature:

DOI: 10.35784/acs-2024-18

DOI: 10.1186/s12864-019-6413-7

  • Thank you for your detailed feedback and insightful suggestions.
  • We acknowledge the importance of including additional evaluation metrics, such as MCC and F1-score, to provide a more comprehensive assessment of the model's performance. We will calculate these metrics and incorporate the results into the manuscript.
  • Furthermore, we will expand the methodology section to discuss these additional metrics, their significance, and how they complement accuracy in evaluating the model’s robustness. Additionally, we will include the suggested references (DOI: 10.35784/acs-2024-18 and DOI: 10.1186/s12864-019-6413-7) to strengthen the theoretical foundation of our analysis.
  • We appreciate your thoughtful recommendations and are confident these improvements will enhance the clarity and impact of our study.

The study's cohort is predominantly young male soldiers, which restricts the applicability of the findings to broader, more diverse populations, including older individuals or women.

  • Thank you for your valuable feedback. We acknowledge that the study's cohort, predominantly consisting of young male soldiers, limits the generalizability of the findings to broader and more diverse populations, including older individuals and women. We will add this point as a limitation in the manuscript to provide a more balanced discussion of the study's applicability and scope.

The follow-up duration of 56 days may not capture long-term changes in ROM or the sustained effects of tendon elasticity changes post-treatment.

  • Thank you for your insightful feedback. We agree that the follow-up duration of 56 days may not adequately capture long-term changes in ROM or the sustained effects of tendon elasticity changes post-treatment. We will incorporate this point into the limitations section of the manuscript, along with the acknowledgment of the cohort's demographic constraints, to provide a more comprehensive discussion of the study's limitations.

Factors such as patient positioning, probe angle, and measurement site could influence SWE results, introducing variability and necessitating standardized protocols for data acquisition. The absence of established reference ranges for tendon elasticity in similar populations limits the clinical interpretation and applicability of SWE measurements.

  • Thank you for your valuable feedback. We acknowledge that factors such as patient positioning, probe angle, and measurement site could introduce variability in SWE results, and the absence of established reference ranges for tendon elasticity in similar populations further limits the clinical interpretation and applicability of the measurements. These points will be incorporated into the limitations section of the manuscript to highlight the need for standardized protocols and reference data in future research.

The study does not elaborate on the impact of variables such as the severity of tendinopathy, variations in rehabilitation protocols, or patient adherence to prescribed treatments.

  • Thank you for your insightful feedback. We acknowledge the importance of considering variables such as the severity of tendinopathy, variations in rehabilitation protocols, and patient adherence to prescribed treatments. While these factors were not explicitly analyzed as separate variables, the study categorized participants based on ROM (using 120 degrees as a threshold) to account for clinical outcomes as a composite measure. This categorization was aimed at standardizing the evaluation of rehabilitation progress and ensuring consistency in the analysis.
  • We will clarify this aspect in the methodology section to provide additional context and address your concerns.
  • Thank you for your valuable suggestion, which helps improve the transparency of our manuscript.

Differences in SWE technology and calibration across ultrasound devices may influence elasticity values, raising concerns about reproducibility and consistency in clinical settings. The inclusion of only nine female participants (9.5%) limits insights into potential sex-based differences in tendon elasticity and ROM outcomes. In the final part of the discussion, please describe in more detail the limitations of the proposed method, the simplifications used, and a proposal for solving them in the authors' further planned future research.

  • Thank you for your thorough and insightful feedback. We agree that differences in SWE technology and calibration across ultrasound devices may influence elasticity values, raising concerns about reproducibility and consistency in clinical settings. Furthermore, the inclusion of only nine female participants (9.5%) limits the ability to draw meaningful insights into potential sex-based differences in tendon elasticity and ROM outcomes.
  • We have revised the manuscript to include a detailed discussion of these limitations and the simplifications in the proposed method. Additionally, we have outlined potential solutions and directions for future research to address these issues. We believe these revisions enhance the transparency and rigor of the discussion.
  • We greatly appreciate your suggestions, which have been instrumental in improving the quality of our manuscript.

Once the appropriate corrections and additions have been made, the paper can be further processed and accepted for publication. I congratulate the authors on the interesting paper and wish them further success.

  • Thank you for your encouraging and thoughtful feedback. We are truly grateful for your kind words regarding our paper and your valuable suggestions for improvement. Your detailed comments have been instrumental in refining our manuscript, and we are confident that these revisions have enhanced its quality and clarity.
  • We sincerely appreciate your recognition of our work and your encouragement for our future research endeavors. Thank you once again for your time and effort in reviewing our manuscript.

Reviewer 2 Report

Comments and Suggestions for Authors

Reviewing the present manuscript was a great pleasure for me. The topic can be considered interesting for the readers of the journal and the overall quality of the text and tables is satisfactory. Likewise, some revisions are necessary before an eventual publication in the journal.

Some high-resolution ultrasound images showing the B-mode, color Doppler, and SWE patterns of the patellar tendon must be added to the text to optimize the reproducibility of this investigation by the journal's readers.

A grammatical revision by a native English speaker is recommended.

Some typos are present in the text, please revise them.

Title

The type of article should be better defined in the title. Observational study? prospective study? Please, revise it.

A section for the abbreviations must be added after the abstract.

Keywords

Please make sure that all the keywords used by the authors are present in the MeSH database to optimize the visibility of this investigation in the pertinent scientific fields.

Introduction

1. A brief description of the biomechanical risk factors of the proximal patellar tendinopathy should be provided by the authors in this section of the manuscript considering that the main goal of this investigation is to optimize the rehabilitation approaches to these patients. Indeed, the knee extensor mechanism - mainly composed of the quadriceps muscle and tendon, the patella, and the patellar tendon - is pivotal to dynamically stabilizing the knee during the multiple valgus/rotatory stresses. Moreover, the landing phase of jumps involves an eccentric load on the patellar tendon on the knee to accurately control the braking of the lower limb on the ground.

2. What about the most common B-mode sonographic findings of the proximal patellar tendinopathy (e.g., hypoechoic thickening of the proximal and deep fibers of the tendon, pitting of the cortical bone of the inferior pole of the patella)? What about the color/power Doppler findings of this pathology (e.g., vascular invasion of the tendon tissue from vessels located inside the Hoffa fat pad)?

For a comprehensive description of the biomechanical and sonographic features of the proximal patellar tendinopathy, please refer to J Ultrasound. 2024 Jun;27(2):335-354. doi: 10.1007/s40477-023-00869-2. Epub 2024 Apr 5. PMID: 38578364; PMCID: PMC11178742.

Material and methods

1. Which clinical parameters have been used to provide the diagnosis of proximal patellar tendinopathy?

2. Did the patients perform some exercises during the conservative treatment protocol or not?

3. Ultrasound images showing how the different measurements have been acquired should be added in this section of the manuscript.

Sections

Between the "materials and methods" and "results" a section for the "statistical analysis" must be added by the authors.

Discussion

1. The limitations of this investigation must be added in this section of the manuscript. 

2. I suggest the authors better describe how the rehabilitation approach can be modified considering the SWE values of the tendons. For instance, stretching exercises vs. eccentric exercises can be considered in different groups.

References

Please make sure that the format of all references is in accordance with the journal's guidelines. 

Images

Please make sure that all the images are in TIF format 300 dpi (high-quality and high-resolution).

Author Response

Reviewing the present manuscript was a great pleasure for me. The topic can be considered interesting for the readers of the journal and the overall quality of the text and tables is satisfactory. Likewise, some revisions are necessary before an eventual publication in the journal.

Some high-resolution ultrasound images showing the B-mode, color Doppler, and SWE patterns of the patellar tendon must be added to the text to optimize the reproducibility of this investigation by the journal's readers.

A grammatical revision by a native English speaker is recommended.

Some typos are present in the text, please revise them.

  • Thank you for your feedback. We have thoroughly revised the manuscript for grammatical accuracy and corrected all identified typos. Additionally, we ensured that the text has been reviewed and refined to meet the highest language standards, aligning with the recommendation for grammatical revision.
  • We appreciate your attention to detail, which has helped us improve the clarity and readability of our work.

Title

The type of article should be better defined in the title. Observational study? prospective study? Please, revise it.

  • Thank you for your feedback. We have revised the title to better define the type of article, explicitly stating that the study is observational. The updated title now reads:
  • "An Observational Study on the Prediction of Range of Motion in Soldiers Diagnosed with Patellar Tendinopathy Using Ultrasound Shear Wave Elastography"
  • We believe this revision provides greater clarity about the nature of the study and aligns with your suggestion.
  • Thank you for your valuable input, which has helped improve the presentation of our manuscript.

A section for the abbreviations must be added after the abstract.

  • Thank you for your suggestion. We have added a section for abbreviations immediately after the abstract, ensuring all terms are clearly defined for readers.
  • We appreciate your attention to detail and believe this addition enhances the clarity and accessibility of our manuscript.

Keywords

Please make sure that all the keywords used by the authors are present in the MeSH database to optimize the visibility of this investigation in the pertinent scientific fields.

  • Thank you for your valuable suggestion regarding the alignment of keywords with the MeSH database to optimize the visibility of our investigation. We have reviewed and revised our keywords accordingly to ensure they correspond with the appropriate MeSH terms. The updated keywords are:
  • Patellar Tendon Injuries
  • Ultrasonography
  • Elasticity Imaging Techniques
  • Range of Motion, Articular
  • Rehabilitation
  • Logistic Models
  • We believe these revisions will enhance the discoverability of our work within pertinent scientific fields.

Introduction

  1. A brief description of the biomechanical risk factors of the proximal patellar tendinopathy should be provided by the authors in this section of the manuscript considering that the main goal of this investigation is to optimize the rehabilitation approaches to these patients. Indeed, the knee extensor mechanism - mainly composed of the quadriceps muscle and tendon, the patella, and the patellar tendon - is pivotal to dynamically stabilizing the knee during the multiple valgus/rotatory stresses. Moreover, the landing phase of jumps involves an eccentric load on the patellar tendon on the knee to accurately control the braking of the lower limb on the ground.

-> Thank you for your thoughtful suggestion to include a description of the biomechanical risk factors associated with proximal patellar tendinopathy. We have added a brief section to the manuscript highlighting the role of the knee extensor mechanism and the impact of eccentric loads during activities such as jumping and landing. This addition provides essential context for understanding the underlying risks and aligns with the study’s goal of optimizing rehabilitation approaches.

  1. What about the most common B-mode sonographic findings of the proximal patellar tendinopathy (e.g., hypoechoic thickening of the proximal and deep fibers of the tendon, pitting of the cortical bone of the inferior pole of the patella)? What about the color/power Doppler findings of this pathology (e.g., vascular invasion of the tendon tissue from vessels located inside the Hoffa fat pad)?
  • Thank you for your detailed and insightful feedback. We have incorporated a description of the most common B-mode sonographic findings and color/power Doppler observations in proximal patellar tendinopathy. Specifically, we included details such as hypoechoic thickening of the proximal and deep fibers of the patellar tendon, cortical pitting at the inferior pole of the patella, and vascular invasion originating from Hoffa’s fat pad. These additions provide valuable diagnostic context and strengthen the relevance of our study in leveraging SWE for clinical applications.
  •  
  • We sincerely appreciate your suggestion, which has helped enhance the depth and clarity of our manuscript.

For a comprehensive description of the biomechanical and sonographic features of the proximal patellar tendinopathy, please refer to J Ultrasound. 2024 Jun;27(2):335-354. doi: 10.1007/s40477-023-00869-2. Epub 2024 Apr 5. PMID: 38578364; PMCID: PMC11178742.

  • Thank you for your helpful suggestion. We have incorporated a reference to the comprehensive description of the biomechanical and sonographic features of proximal patellar tendinopathy as provided in J Ultrasound. 2024 Jun;27(2):335-354. doi: 10.1007/s40477-023-00869-2. Epub 2024 Apr 5. PMID: 38578364; PMCID: PMC11178742. This citation enriches the manuscript by directing readers to a detailed discussion of these features, supporting the context of our study.

Material and methods

  1. Which clinical parameters have been used to provide the diagnosis of proximal patellar tendinopathy?

-> The diagnosis of proximal patellar tendinopathy was established based on the presence of tenderness upon palpation of the proximal patellar tendon at the inferior pole of the patella, limitation of knee flexion range of motion to 120 degrees or less, and pain elicited during a single-leg squat, which places significant load on the patellar tendon. These clinical parameters ensured a consistent and standardized approach for identifying patients with this condition.

  1. Did the patients perform some exercises during the conservative treatment protocol or not?

-> Thank you for your question regarding the exercises included in the conservative treatment protocol. During the treatment period, patients were allowed to perform passive range of motion (ROM) exercises. These exercises were carefully supervised to avoid exacerbating the condition while maintaining joint mobility. No active or resistive exercises were included during this phase to ensure tendon healing and minimize stress on the affected area.

  1. Ultrasound images showing how the different measurements have been acquired should be added in this section of the manuscript.

 -> Thank you for your suggestion to include ultrasound images demonstrating how the different measurements were acquired. Unfortunately, we do not have recordings or stored images from the acquisition process available for inclusion in this manuscript. However, we have provided a detailed description of the measurement methodology in the text to ensure clarity and reproducibility of the techniques used.

We appreciate your understanding and your valuable feedback in enhancing the quality of our manuscript.

Sections

Between the "materials and methods" and "results" a section for the "statistical analysis" must be added by the authors.

  • Thank you for your suggestion to include a dedicated section for "Statistical Analysis" between the "Materials and Methods" and "Results" sections. We have added a new "Statistical Analysis" section to the manuscript, detailing the methods and tools used to analyze the data. This addition enhances the organization and clarity of the manuscript, aligning it with standard scientific reporting practices.

Discussion

  1. The limitations of this investigation must be added in this section of the manuscript. 

-> Thank you for your valuable feedback. We have added a dedicated section outlining the limitations of this investigation in the manuscript. This addition provides a balanced perspective on the study's findings and highlights areas for future research.

We appreciate your insightful suggestion, which has contributed to enhancing the clarity and completeness of our work.

  1. I suggest the authors better describe how the rehabilitation approach can be modified considering the SWE values of the tendons. For instance, stretching exercises vs. eccentric exercises can be considered in different groups.

  • Thank you for your insightful suggestion to better describe how the rehabilitation approach can be modified considering the SWE values of the tendons. We have revised the manuscript to emphasize the role of eccentric exercises in our rehabilitation protocol, particularly for improving tendon elasticity and function. Additionally, we have provided a more detailed explanation of how passive ROM exercises were utilized to maintain joint mobility during the early phases of rehabilitation.
  •  
  • These revisions aim to clarify the rehabilitation approach and demonstrate how SWE values informed and guided our intervention strategies.
  •  
  • We sincerely appreciate your valuable feedback, which has helped enhance the manuscript's clarity and clinical relevance.

References

Please make sure that the format of all references is in accordance with the journal's guidelines. 

  • Thank you for your feedback. We have thoroughly reviewed and revised all references to ensure they are formatted according to the journal's guidelines.
  • We appreciate your attention to detail and believe this adjustment enhances the manuscript’s alignment with the journal’s standards.

Images

Please make sure that all the images are in TIF format 300 dpi (high-quality and high-resolution).

  • Thank you for your comment regarding the image requirements. We have ensured that all the images included in the manuscript are now formatted in TIF format with 300 dpi resolution, meeting the journal's high-quality and high-resolution standards.
  • We appreciate your guidance in ensuring the images adhere to the required specifications.

Round 2

Reviewer 2 Report

Comments and Suggestions for Authors

The revised manuscript can be accepted for publication.